# Study on Improving the Water Quality of Drilling Mud Using Industrial Waste Residue

**Yuyue Jia [1], Huayang Li [1], Yun Meng [1] and Nan Li [2,\*]**

[1]  School of Civil Engineering, Qingdao University of Technology, Qingdao 266033, China
[2]  School of Mechanical and Automotive Engineering, Qingdao University of Technology, Qingdao 266033, China
\*   Correspondence: linan@qut.edu.cn; Tel.: +86-13488270721

**Abstract:** Drilling waste has a significant effect on the water quality and the aquatic environment. Therefore, the harmless solidification treatment of drilling waste mud is important in terms of protecting the water environment. However, traditional chemical solidification methods have some problems, such as wide area, high cost, and secondary pollution. Therefore, the improvement of the mechanical performance on treating the drilling waste is important, as it can reduce and minimize the effect of drilling waste on the water environment. To this end, in this study, the industrial waste residue is used as the main component of the curing agent and a harmless curing scheme is designed. According to the orthogonal design method, an unconfined compressive strength test is carried out on the solidified soil of drilling waste mud. The strength, stress-strain relationship, and deformation modulus of the solidified soil are discussed according to the curing age, the number of freeze-thaw cycles, and the proportion of the solidifying agent. The test results show that the unconfined compressive strength of solidified soil samples under different proportions increases with the increase of the curing age and decreases with the increase of the freeze-thaw cycles. With the Increase of the curing age, the solidified soil gradually changes from plasticity to brittleness, and its deformation modulus $E_{50}$ also increases gradually.

**Keywords:** water environment; drilling waste mud; curing agent; mechanical properties; freeze-thaw cycle; industrial waste residue

## 1. Introduction

Oil and natural gas are important energy resources globally. During the drilling construction work of oil and gas, the production of waste mud is inevitable. Such drilling waste is a muddy substance used to lubricate, cool the drill tip, and balance the downhole pressure during oilfield drilling. The main components are bentonite, brine, and various chemical treatment agents. Such drilling waste generated during the drilling work in the seas will have a significant effect on water quality and the environment. [1].

Due to the various pollutions produced during drilling, arbitrary discharge tends to cause serious damage to the surrounding environment (e.g., water environment) and human health. Accordingly, increasing attention to environmental protection has been drawn, particularly in China. As such, the investigation for the appropriate treatment of waste mud from drilling has also become a priority in the industrial field. Jamrozik et al. [2] utilized environmentally friendly chemicals, waste reuse, and recycling technologies to achieve waste management, fluid and solid control of drilling waste, thereby improving the economic benefits and minimizing the impact of drilling on the environment. Pereira et al. [3] mentioned in their research that only bioremediation and curing have proven easy to use on a large scale. However, the bioremediation process lasts a long time, and it is difficult to achieve high biodegradable efficiency in an uncontrolled environment. At present, the most widely used approach is the harmless solidification treatment, by adding

the prepared solidifying agent to the mud pool for in-situ solidification [4]. However, disadvantages are noticeable, such as the significant space occupation, high cost of curing agents, production of secondary pollution, and challenge of reusing waste.

The previous study has indicated that using industrial waste residue as the raw material to prepare curing agents for filling roadbeds and improving soil quality could be an effective solution. This can not only treat the waste industrial waste harmlessly, but also reduce the material cost in engineering construction, which is in line with the trend of environmental-friendly resource utilization. For example, Boutammine et al. [5] proposed a method combining stabilization/solidification (S/S) and biological treatment. Portland cement is used as a hydraulic binder. Some of the cement consumption was partially replaced by two additives, compost and bentonite. The results showed that adding 2%, 4%, and 6% additives reduced the total petroleum hydrocarbons (TPH) content of bentonite to 29.47%, 33.96%, and 37.21%, and the TPH content of compost to 53.17%, 61.51%, and 64.43%. Cheng et al. [6] combined the red mud and blast furnace slag with drilling waste mud, from which a new type of cementitious material was produced which had a compressive strength of up to 16.7 MPa. He et al. [7] utilized three industrial by-products i.e., soda residue (SR), carbide slag (CS), and ground granulated blast slag (GGBS), to solidify dredged soil at a high moisture content; their experimental result showed that the unconfined compressive strength of the specimen with an initial water content of 110% increased with a higher portion of ore powder content introduced, in which the optimal mixing fractions of SR and CS are 35% and 6%, respectively. Cheng et al. [8] used cement and industrial waste slag to solidify the drilling waste mud in the Changqing oilfield, and they found that the unconfined compressive strength of the four ratio-optimized solidified soils would decrease with the increase of freeze-thaw cycles, and an increase in phosphogypsum content could effectively reduce the strength loss caused by the freeze-thaw process. Gao et al. [9] investigated the uniaxial compressive strength of cement, fly ash, ultrafine slag powder and plant ash after the solidification treatment of waste engineering mud via in-house laboratory tests. In addition to that, they carried out the blade shear and field loading tests, putting forward a prediction formula for the ultimate bearing capacity of the in-situ solidified soil [10].

Petroleum coke desulfuration residues are the combustion product of CFB boilers in petroleum refineries. They belong to industrial waste. Petroleum coke desulfuration residue, an industrial waste in the petroleum industry, can be used as a material to solidify drilling mud, which can realize the recycling of waste. Following the above context and based on the traditional cement solidification method, in this study, a drilling waste mud solidification treatment is carried out by using a small cement content, petroleum coke desulfuration residues, and fly ash to investigate the potential improvement in the mechanical properties and durability of the mud, eventually aiming for the triple utilization of drilling waste mud, petroleum coke, and fly ash. Correspondingly, (a) the unconfined compressive strength of the solidified soil, (b) the stress-strain relationship and deformation modulus of the solidified soil, and (c) the weakening effect of a freeze-thaw cycle on the strength of solidified soil will be analyzed in the study.

## 2. Materials and Methods

### 2.1. Test Materials and Methods

(1) Waste Drilling Mud

The waste drilling mud for the test was collected from a drilling site in Dongying Shengli Oilfield, Shandong Province. It seems to be very hard and dark brown in color after natural drying and air drying, as shown in Figure 1. After that, it was ground and dried to obtain the waste drilling mud powder (Figure 2). The basic physical and chemical properties of the mud are shown in Table 1. The chemical composition and heavy metal ion content of waste drilling mud are analyzed by an X-ray fluorescence spectrometer, as shown in Tables 2 and 3, respectively.

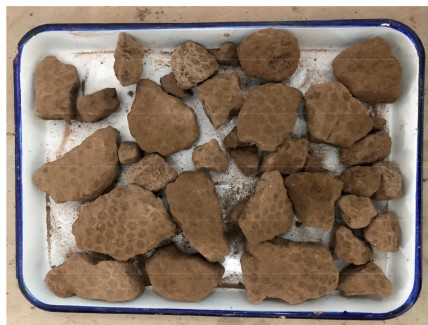

**Figure 1.** Original mud consolidated soil sample.

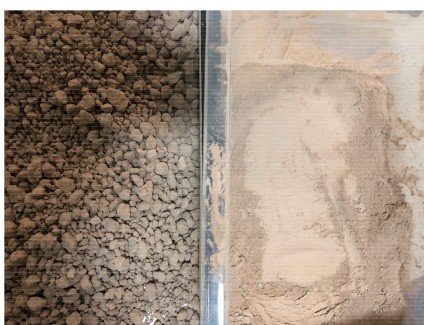

**Figure 2.** Dry powder.

**Table 1.** Physical properties of waste drilling fluid.

| Moisture Content (%) | Density (g/cm$^3$) | Liquid Limit (%) | Plastic Limit (%) | Liquid Index | Plasticity Index | Optimum Water Content (%) | Maximum Dry Density (g/cm$^3$) |
|---|---|---|---|---|---|---|---|
| 48.4 | 1.35 | 45.3 | 23.2 | 1.14 | 22.1 | 19.3 | 1.653 |

**Table 2.** Chemical composition of drilling mud.

| Chemical Composition (wt. %) | | | | | | | | | | |
|---|---|---|---|---|---|---|---|---|---|---|
| Na$_2$O | MgO | Al$_2$O$_3$ | SiO$_2$ | P$_2$O$_5$ | SO$_3$ | Cl | K$_2$O | CaO | Fe$_2$O$_3$ | Loss |
| 5.135 | 2.675 | 13.415 | 42.465 | 0.136 | 1.460 | 3.425 | 2.210 | 9.118 | 4.217 | 17.310 |

**Table 3.** Analysis of heavy components of drilling waste mud.

| Heavy Metal Content (wt. %) | | | | | | | | |
|---|---|---|---|---|---|---|---|---|
| TiO$_2$ | MnO | ZrO$_2$ | CuO | ZnO | Rb$_2$O | SrO | Y$_2$O$_3$ | BaO |
| 0.607 | 0.061 | 0.020 | - | 0.015 | 0.008 | 0.045 | 0.003 | 0.052 |

(2)  Curing Materials

According to the characteristics of waste drilling mud, combined with engineering practice and geographical conditions, petroleum coke desulfuration residues (PCDR), cement (OPC), and fly ash (FA) are selected as the main components in this study. Desulphurized petroleum coke slag is a by-product of Sinopec Qingdao Refining and Chemical Co., Ltd.,( Qingdao, China) which is mixed with quicklime powder and sprayed into the fluidized bed boiler for desulfurization treatment. It is mainly composed of CaO and CaSO4, showing a light yellow color. The cement is P.O. 42.5 cement produced by the Shanshui cement factory in Shandong Province. The fly ash is selected from the combustion solid waste of a thermal power plant in Qingdao, Shandong Province. The three cured materials were analyzed by X-ray fluorescence spectroscopy (XRF), and the results are shown in Table 4.

**Table 4.** XRF composition analysis of PCDR, OPC, and FA in the current study.

| Materials | SiO$_2$ | Al$_2$O$_3$ | Fe$_2$O$_3$ | CaO | MgO | K$_2$O | Na$_2$O | SO$_3$ | TiO$_2$ | Other |
|-----------|---------|-------------|-------------|-------|------|--------|---------|--------|---------|-------|
| OPC | 22.41 | 5.63 | 3.40 | 62.46 | 1.23 | 0.24 | 0.32 | 2.80 | 0.92 | 0.59 |
| FA | 35.91 | 29.03 | 4.50 | 13.97 | 3.05 | 0.62 | 0.38 | 6.74 | - | 5.80 |
| PCDR | 2.01 | 1.03 | 0.37 | 51.77 | 2.55 | - | - | 29.97 | - | 12.30 |

*2.2. Test Method*

2.2.1. Preparation of Solidified Soil

Three components of dry drilling waste mud powder and curing agent: desulphurized petroleum coke slag, cement, and fly ash, are sealed and stored in a sealed bag to prevent moisture from entering the air. Take out the dry soil and water with an electronic scale and mix them with an electric mixer for 5 min. The well mixed mud is used as the original mud for the test. Weigh three curing agent components, according to the set proportion, and add them into the original mud three times. After each addition, use an electric mixer to fully mix for 2 min. The purpose of this is to ensure that the moisture content of the mixed mud is 50%. After, all the solidified soil was mixed and dumped into a plastic box. It was placed indoors for 1 day, to further reduce the moisture content of solidified soil and improve the success rate of sample preparation.

2.2.2. Proportioning Test of Curing Agent

The orthogonal test scheme design of three factors and three levels is selected, as shown in Table 5.

**Table 5.** Orthogonal experiment scheme.

| Proportion Number | Cement/% | Desulphurized Petroleum Coke Slag/% | Fly Ash /% |
|-------------------|----------|-------------------------------------|------------|
| O1P1F1 | 8 | 15 | 12 |
| O1P2F2 | 8 | 22.5 | 16 |
| O1P3F3 | 8 | 30 | 20 |
| O2P1F2 | 12 | 15 | 16 |
| O2P2F3 | 12 | 22.5 | 20 |
| O2P3F1 | 12 | 30 | 12 |
| O3P1F3 | 16 | 15 | 20 |
| O3P2F1 | 16 | 22.5 | 12 |
| O3P3F2 | 16 | 30 | 16 |

Note: the figure in the table is the ratio of each component of the curing agent to the dry weight of test drilling mud.

2.2.3. Mechanical Property Test

(1) Unconfined compressive strength test

The unconfined compressive strength test adopts a cylindrical specimen with a diameter of 39.1 mm and a height of 80 mm. Three parallel samples were made in each group, and the quality was weighed by the electronic scale. If the mass difference of the three parallel samples was more than 5 g, it would be remade. The group and the curing age were marked on the side of the weighing qualified sample, and then the marked sample was placed in the standard curing room for maintenance, in which the curing temperature was 20 ± 2 °C and the curing humidity was 97%.

(2) Simulation of the freeze-thaw cycle

The sample cured for 28 days was immersed in water for 1 day to saturate the sample and then the water was drained for 1 h. The samples were wrapped with sealed plastic film and put into the freeze-thaw cycle test box. Then the unconfined compressive strength test was carried out for 0, 2, 4, 6, 8, and 10 freeze-thaw cycles. A single freeze-thaw cycle

lasts for 24 h, and the specific process is as follows: 12 h at $-15\ ^\circ$C, and then 12 h at 15 $^\circ$C, which is a freeze-thaw cycle.

## 3. Results and Discussion

### 3.1. Orthogonal Test Results Analysis

3.1.1. Macroscopic Analysis

Table 6 of the orthogonal test results shows that, when treating waste drilling mud with 50% water for a curing period of 3 days, 7 days, and 28 days, the strength of the sample with the O1P2F2 ratio seems to be high and the corresponding strength values are 288.2 kPa, 577.5 kPa, and 1792.3 kPa, respectively. At this time, the corresponding proportion of the curing agent is as follows: 8% of cement, 22.5% of desulphurized petroleum coke slag, and 16% of fly ash. It is also observed that the strength of the samples with the proportion of O3P3F2 (when the curing age is 3 days and 7 days) and O3P2F1 (when the curing age is 28 days) is the lowest, and the corresponding strengths are 160.1 kPa, 304.8 kPa, and 995.9 kPa, respectively.

**Table 6.** Results of orthogonal experiment.

| Proportion Number | Cement (A) | Desulphurized Petroleum Coke Slag (B) | Fly Ash (C) | Unconfined Compressive Strength (kPa) | | |
|---|---|---|---|---|---|---|
| | | | | 3 Days | 7 Days | 28 Days |
| O1P1F1 | 1 (8%) | 1 (15%) | 1 (12%) | 253.1 | 457.7 | 1790.2 |
| O1P2F2 | 1 | 2 (22.5%) | 2 (16%) | 288.2 | 577.5 | 1792.3 |
| O1P3F3 | 1 | 3 (30%) | 3 (20%) | 205.6 | 394.6 | 1721.6 |
| O2P1F2 | 2 (12%) | 1 | 2 | 257.2 | 532.0 | 1726.6 |
| O2P2F3 | 2 | 2 | 3 | 249.0 | 423.6 | 1689.0 |
| O2P3F1 | 2 | 3 | 1 | 241.8 | 432.9 | 1292.3 |
| O3P1F3 | 3 (16%) | 1 | 3 | 185.9 | 370.9 | 1365.6 |
| O3P2F1 | 3 | 2 | 1 | 231.4 | 375.0 | 995.9 |
| O3P3F2 | 3 | 3 | 2 | 160.1 | 304.8 | 1044.4 |

3.1.2. Range Analysis

The range analysis of the orthogonal test results is shown in Table 7. It can be analyzed that the primary and secondary relationship of the factors affecting the unconfined compressive strength of solidified soil for 3 days curing age is as follows: cement > desulphurized petroleum coke slag > fly ash. The primary and secondary relationship of the factors affecting the unconfined compressive strength under 7-day curing age is cement > fly ash > desulphurized petroleum coke slag. When the curing age is 28 days, the primary and secondary relationship of the factors affecting the unconfined compressive strength of the solidified soil is the same as that of 3 days curing age, that is, cement > desulphurized petroleum coke slag > fly ash. Under each curing age, the optimal level combination of the three factors is O1P2F2; that is, under the existing curing agent level, the cement content is low, the desulphurized petroleum coke slag and fly ash content is moderate, and the strength is the highest.

The reason for the above phenomenon may be that the curing effect of the curing agent on mud can be divided into long-term effects and short-term effects, in which the former is mainly a hydration reaction, while the latter includes water absorption, flocculation, and a carbonization reaction [11]. In this paper, after the cement and mud are mixed and stirred, the cement mineral undergoes hydrolysis and hydration reactions, and the calcium hydroxide is decomposed. It can also promote the formation of other hydrates in the soil. The skeleton effect of calcium hydroxide is obvious, so cement has the greatest influence at each curing age [12]. However, the effect of desulphurized petroleum coke residue is more obvious at 7 days and 28 days, because its content is larger, and its main composition is quicklime and calcium sulfate; it reacts quickly with the free water and pore water in the mud. A series of physical and chemical reactions occur after lime and mud are mixed and stirred, mainly including an ion exchange reaction, calcium hydroxide crystallization reaction, pozzolanic reaction, etc. Ion exchange can thin the colloid adsorption layer and improve the wet slump of mud so that the improved mud

has initial water stability. The calcium hydroxide crystallization reaction is to improve the strength of soil by forming crystals. The pozzolanic reaction is that the mud is dissociated under the alkaline excitation of lime, and then generates hydrated calcium silicate, calcium aluminate, and other cement in the water environment, thus improving the stability and strength of the mud. Calcium sulfate reacts with calcium aluminate hydrate in cement hydration products to form ettringite, which is characterized by volume expansion. It can effectively fill the pores and improve the mud strength, but it takes some time for the strength of the hydration products to increase. [13].

**Table 7.** Orthogonal extreme difference analysis.

| | **3-Day UCS/kPa** | | | **7-Day UCS/kPa** | | | **28-Day UCS/kPa** | | |
|---|---|---|---|---|---|---|---|---|---|
| | **OPC (A)** | **PCDR (B)** | **FA (C)** | **OPC (A)** | **PCDR (B)** | **FA (C)** | **OPC (A)** | **PCDR (B)** | **FA (C)** |
| K1 | 746.9 | 696.2 | 655.0 | 1429.8 | 1360.6 | 1195.4 | 5304.1 | 4882.4 | 4126.9 |
| K2 | 748.0 | 768.6 | 776.8 | 1388.5 | 1376.1 | 1484.5 | 4707.9 | 4477.2 | 4514.8 |
| K3 | 577.4 | 607.5 | 640.5 | 1050.7 | 1132.3 | 1189.1 | 3405.9 | 4058.3 | 4776.2 |
| K1 | 249.0 | 232.1 | 218.3 | 476.6 | 453.5 | 398.5 | 1768.0 | 1627.5 | 1375.6 |
| K2 | 249.3 | 256.2 | 258.9 | 462.8 | 458.7 | 494.8 | 1569.3 | 1492.4 | 1504.9 |
| K3 | 192.5 | 210.8 | 205.2 | 350.2 | 377.4 | 396.4 | 1135.3 | 1352.8 | 1592.1 |
| R | 56.9 | 45.4 | 53.7 | 126.4 | 81.3 | 98.5 | 632.7 | 274.7 | 216.4 |
| Influence factor | A > C > B | | | A > B > C | | | A > B > C | | |

Note: $K_i$ represents the sum of the unconfined compressive strength corresponding to the first level of each factor under a certain nursing age, $i$ is the horizontal number of each factor. $\overline{K_i}$ represents the mean of the sum of unconfined compressive strength to the horizontal number, $\overline{K_i} = K_i/3$ is the difference between the maximum and minimum $K_i$ of each factor, that is, the range. The larger the $R$, the more obvious the influence of this factor on the unconfined compressive strength.

According to the principle of range analysis, the changing trend chart of each factor is drawn, as shown in Figure 3. It can be seen that, when the curing time is 3 days, the effect of cement content at the first level and the second level on the unconfined compressive strength is almost the same, but when it is increased to the third level, the strength will decrease. However, the influence trend of desulphurization petroleum coke slag and fly ash is the same, and the strength is the highest at the second level, the second at the first level, and the lowest at the third level. When the curing time is 7 days, the unconfined compressive strength of cement at the first level is slightly higher than that at the second level, and It is still the lowest at the third level. However, the strength of desulphurized petroleum coke slag still reached the peak at the second level, but the strength difference with the first level was small, and the lowest at the third level, and the changing trend of fly ash was the same as that at 3 days. When the curing time is 28 days, the changing trend of cement and desulfurized petroleum coke slag is almost the same, which is the first level > the second level > the third level, while the fly ash is just the opposite, and the changing trend is the third level > the second level > the first level.

### 3.1.3. Analysis of Variance

According to the results of the previous orthogonal test, the analysis of variance is shown in Table 8. It can be seen that the F value of 3 days curing period is $F_A > F_B > F_C$, indicating that cement has the greatest influence on the strength of 3 days, followed by desulfurized petroleum coke slag and fly ash, and the influence of three factors on unconfined compressive strength is not significant. The F value of 7 days is $F_A > F_B > F_C$, so cement has the greatest influence on the strength of 7 days, followed by desulphurized petroleum coke slag and fly ash, but the difference is reduced, and the influence of three factors on strength is still not significant. Under the curing age of 28 days, the order of

the F value is still $F_A > F_B > F_C$, so cement has the greatest influence on the strength of 7 days, followed by desulphurized petroleum coke slag and fly ash. At this time, the F value of cement is 10.488, which is quite different from that of the other two curing agents, which proves that, with the increase of curing age, the effect of cement is more and more significant.

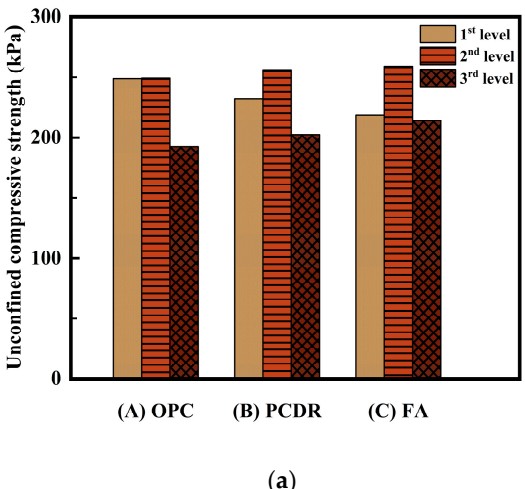

**(a)**

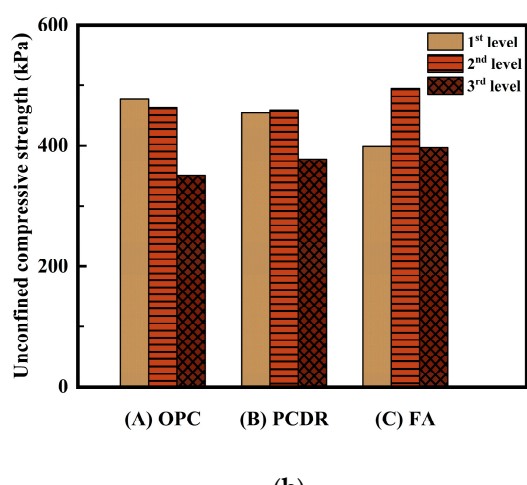

**(b)**

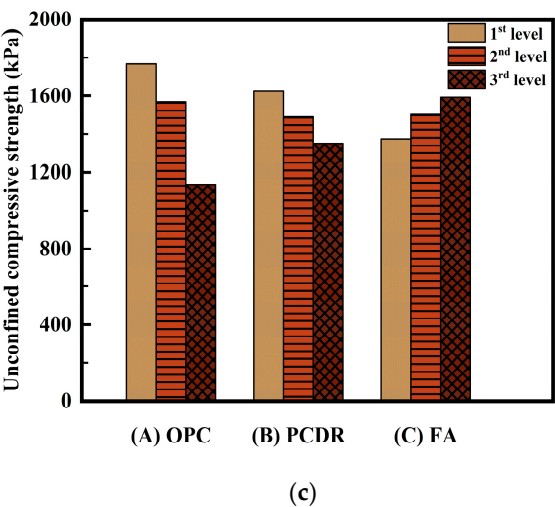

**(c)**

**Figure 3.** Extremely bad trend chart of orthogonal test results when the initial water content of mud is 50%. (**a**) 3 days curing age. (**b**) 7 days curing age. (**c**) 28 days curing age.

**Table 8.** Analysis of variance of orthogonal test results when the initial water content of mud is 50%.

| Variance Source | Sum of Squares of Column Differences | Degree of Freedom | The Value of F | Significant Level |
|---|---|---|---|---|
| | | Intensity (3 days)/kPa | | |
| A (OPC) | 6426.2 | 2 | 4.275 | Not significant |
| B (PCDR) | 2304.6 | 2 | 1.533 | Not significant |
| C (FA) | 318.9 | 2 | 0.212 | Not significant |
| E (Error) | 1503.1 | 2 | | |
| Total | 10552.8 | 8 | | |
| | | Intensity (7 days)/kPa | | |
| A (OPC) | 28,836.8 | 2 | 5.459 | Not significant |
| B (PCDR) | 18,001.7 | 2 | 3.408 | Not significant |
| C (FA) | 9881.2 | 2 | 1.871 | Not significant |
| E (Error) | 5282.6 | 2 | | |
| Total | 62,002.3 | 8 | | |
| | | Intensity (28 days)/kPa | | |
| A (OPC) | 628,202.4 | 2 | 10.488 | Significant |
| B (PCDR) | 139,570.6 | 2 | 2.330 | Not significant |
| C (FA) | 26,672.7 | 2 | 0.445 | Not significant |
| E (Error) | 59,896.6 | 2 | | |
| Total | 854,342.3 | 8 | | |

### 3.2. Effect of Curing Age on Unconfined Compressive Strength

The variation of the unconfined compressive strength of solidified soil samples with waste drilling mud with curing age is shown in Figure 4. It can be seen that, after the mud is solidified, the strength of all samples increases with the increase of curing age. Taking the three proportions of the best, moderate, and worst strength properties as typical examples. The 3 days strength of the solidified soil with the ratio scheme of O1P2F2 was 288.2 kPa, and the 7 days and 28 days strengths increased to 577.5 kPa and 1972.3 kPa, respectively, increasing by 100% and 522%. The 3 days strength of the solidified soil with the ratio scheme of O2P2F3 was 249.0 kPa, and the 7 days and 28 days strengths increased to 423.6 kPa and 1689.0 kPa, respectively, increasing by 70% and 578%. The 3 days strength of the solidified soil with the ratio scheme of O3P2F1 was 231.4 kPa, and the 7 days and 28 days strengths increased to 375.0 kPa and 995.9 kPa, respectively, increasing by 62% and 330%.

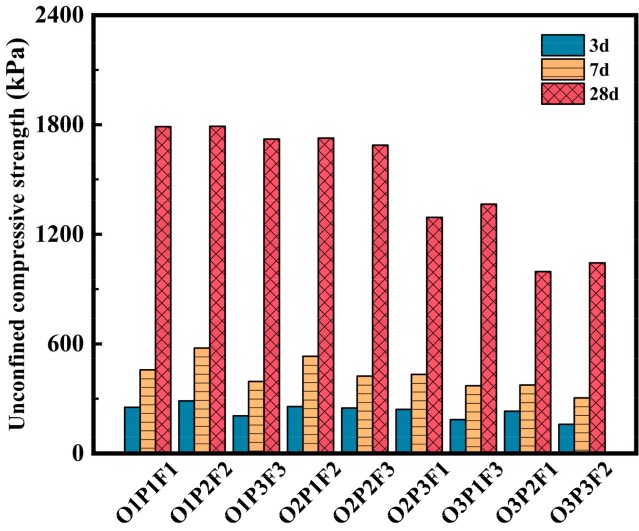

**Figure 4.** Unconfined compressive strength of solidified soil at different curing ages.

It can be concluded that the unconfined compressive strength of solidified soil increases with the increase of curing age, gradually stabilizes after 28 days, and does not change the law of development because of the different proportioning schemes, and the growth rate is roughly the same. Compared with 3 days strength, the growth rates of 7 days strength and 28 days strength are about 62–107% and 330–738%, respectively. This is similar to the conclusion drawn by Yang Jun. [14] studied the strength of cement-modified soil. The reason may be that, with the addition of curing agent, a series of chemical reactions take place in the mud, such as the pozzolanic reaction, ion exchange, and so on, resulting in the formation of strong cement. With the increase in curing age, the internal reaction tends to be complete, which leads to the slow growth of strength in the later stage.

### 3.3. Effect of the Freeze-Thaw Cycle on Unconfined Compressive Strength

Based on the preliminary screening of nine kinds of curing agent ratios, three optimum ratios of O1P1F1, O1P2F2, and O2P1F2 are obtained, and the unconfined compressive strength test of solidified soil after the freeze-thaw cycle is carried out. The test results are shown in Figure 5. With the increase in the number of freeze-thaw cycles, the strength of solidified soil with three ratios decreased gradually. After 10 freeze-thaw cycles, the compressive strength of O1P1F1 solidified soil is 16.7% lower than that of the sample without the freeze-thaw cycle, which is about 1484.3 kPa. After 10 freeze-thaw cycles, the compressive strength of O1P2F2 solidified soil is 21.9% lower than that of the sample without the freeze-thaw cycle, which is about 1401.7 kPa. After 10 freeze-thaw cycles, the compressive strength of the O2P1F2 solidified soil sample is 17.9% lower than that of the sample without the freeze-thaw cycle, which is about 1421.8 kPa.

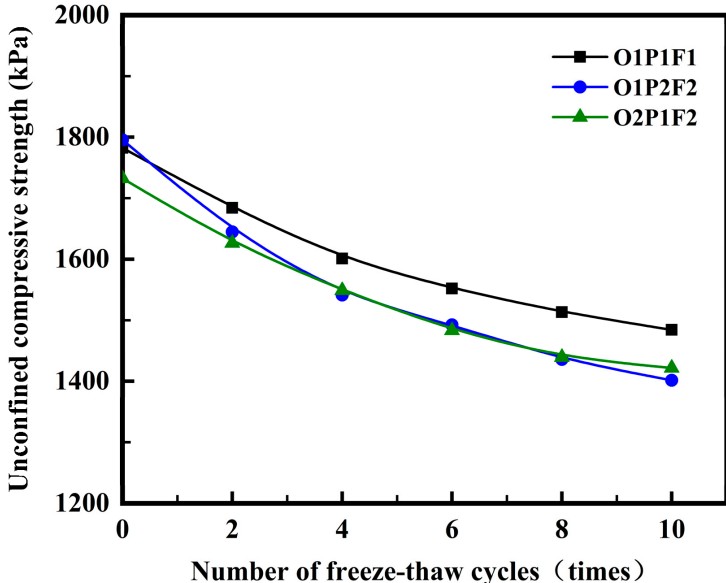

**Figure 5.** Unconfined compressive strength of specimens under freeze-thaw cycles.

At present, the unified understanding of the mechanism of freeze-thaw damage of solidified soil is that the volume expansion of pore water in solidified soil in the process of freezing and crystallization leads to the destruction of the solidified soil structure, which reduces the strength of solidified soil. During the freezing process of solidified soil, the temperature of the surface of the sample is lower than that of the interior of the sample, and the moisture inside the sample migrates and accumulates to the surface, which makes the volume expansion effect of the surface of the sample more significant, and the degree of damage is higher. During the melting process of the sample, the temperature of the surface of the sample is higher than that of the interior of the sample, and the water melted on the surface of the sample migrates to the interior, resulting in the migration and accumulation of soluble matter in the sample, which affects the internal structure of the sample. In

the process of freezing and thawing repeatedly, the soil structure is gradually destroyed, resulting in a significant decrease in strength [8].

In addition, the decrease in strength of all samples is the largest when the freeze-thaw cycles are 0–4 times. It is noted that the strength is decreased during the 4th cycle to the 8th cycle and becomes stable after the 8th cycle. After four freeze-thaw cycles, the compressive strength of O1P1F1 solidified soil samples decreased by 11.3%, accounting for 67.7% of the total attenuation of 10 freeze-thaw cycles, and the strength was about 1601.2 kPa. After eight freeze-thaw cycles, the compressive strength decreased by 15.1%, accounting for 90.2% of the total attenuation of 10 freeze-thaw cycles, and the strength was about 1513.4 kPa. After four freeze-thaw cycles, the compressive strength of O1P2F2 solidified soil samples decreased by 14.2%, accounting for 64.6% of the total attenuation of 10 freeze-thaw cycles, and the strength was about 1644.9 kPa. After eight freeze-thaw cycles, the compressive strength decreased by 20.0%, accounting for 91.3% of the total attenuation of 10 freeze-thaw cycles, and the strength was about 1435.8 kPa. After four freeze-thaw cycles, the compressive strength of O2P1F2 solidified soil samples decreased by 59.1%, accounting for 59.1% of the total attenuation of 10 freeze-thaw cycles, and the strength was about 1548.9 kPa. After eight freeze-thaw cycles, the compressive strength decreased by 16.9%, accounting for 94.3% of the total attenuation of 10 freeze-thaw cycles, and the strength was about 1439.4 kPa.

The reason is that the water in the soil does not freeze and expand as expected, which oppresses the dislocation of soil particles, produces microcracks, and destroys the whole structure of solidified soil samples. Therefore, after the first two freeze-thaw cycles, the strength of the solidified soil decreases the most; after six freeze-thaw cycles, there are a considerable number of microcracks, which provides enough space for the subsequent icing expansion, and the subsequent single freezing damage is weakened, so the strength of the solidified soil decreases obviously and tends to be stable gradually [15].

*3.4. Stress-Strain Relationship of Solidified Soil*

Because the stress-strain curves of nine groups of solidified soil samples under the same curing age are roughly the same, three typical samples with the highest strength, moderate strength, and lowest strength are selected as the analysis object, as shown in Figure 6. When the curing age is 3 days, the proportions of the highest, moderate, and lowest strength are O1P2F2, O2P3F1, and O3P3F2, respectively. It can be seen that the stress-strain curves of the three samples increase at first and then decrease, and the samples have a certain plasticity. When the curing age is 7 days, the proportions of the highest, moderate, and lowest strength are O1P2F2, O2P2F3, and O3P3F2, respectively. The stress-strain curves of the three samples increase at first and then tend to smooth until the specimen is destroyed when the plasticity of the sample attenuates, and the brittleness is gradually obvious. When the curing age is 28 days, the proportions of the highest, moderate, and lowest strength are O1P2F2, O2P2F3, and O3P2F1, respectively. The stress-strain curves of the three samples increase to the failure of the samples when the specimens show complete brittleness and plasticity disappears.

In addition, when the curing age is 3 days, the strain range of sample disintegration failure is concentrated at 2.0–3.0%. The strain of the sample reduced to 1.5–2.0% during disintegration and failure during the 7-day curing age. When cured for 28 days, the strain range of sample disintegration failure is reduced to 0.75–1.5%. It can be seen that, with the increase of the curing age, the strain of disintegration failure of the sample is smaller, and the stress-strain curve is steeper, which proves that the brittleness becomes stronger and the plasticity disappears gradually.

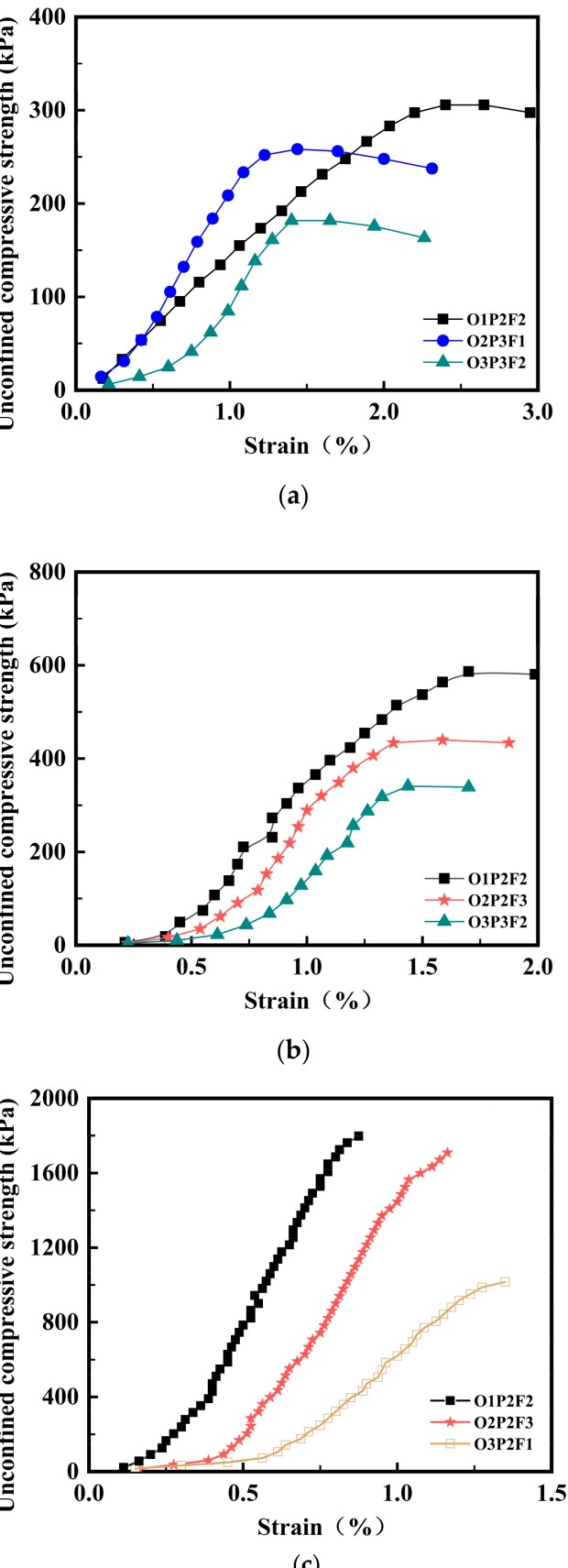

**Figure 6.** Stress-strain curve of solidified soil. (**a**) 3 days curing age. (**b**) 7 days curing age. (**c**) 28 days curing age.

*3.5. Deformation Modulus of Solidified Soil*

$E_{50}$ refers to the 1/2 cutting modulus of the solidified soil strain of drilling waste mud, which is essentially the deformation modulus of solidified soil [16]. The larger the $E_{50}$, the stronger the hardness and brittleness of the sample and the better the compression resistance. The formula is as follows:

$$E_{50} = \frac{2\sigma_{1/2}}{\varepsilon_f} \tag{1}$$

where: $\varepsilon_f$—yield strain, the strain value corresponding to the peak strength; $\sigma_{1/2}$—the stress value when the compressive strain reaches half of $\varepsilon_f$.

The deformation modulus of solidified soil with drilling waste mud is shown in Figure 7. It can be seen that the deformation modulus of each group of solidified soils roughly increases with the increase of curing age, which indicates that the increase in curing age can enhance the deformation resistance of the sample. Among all the samples, when the curing age is 7 days, the deformation modulus of the sample with O3P3F2 is the lowest, which is only 85.9 Mpa, indicating that the sample with this scheme has the weakest ability to resist deformation. When the curing age is 3 days, the deformation modulus of the sample with the O2P3F1 ratio is the highest, which is about 210.0 Mpa. When the curing age is 7 days, the deformation modulus of the sample with the O1P2F2 ratio is the highest, which is about 351.1 Mpa. When the curing age is 28 days, the deformation modulus of the sample with the proportion of O1P2F2 is the highest, which is about 1335.6 Mpa. This shows that the three groups of samples have the strongest ability to resist deformation under the condition of their respective curing age.

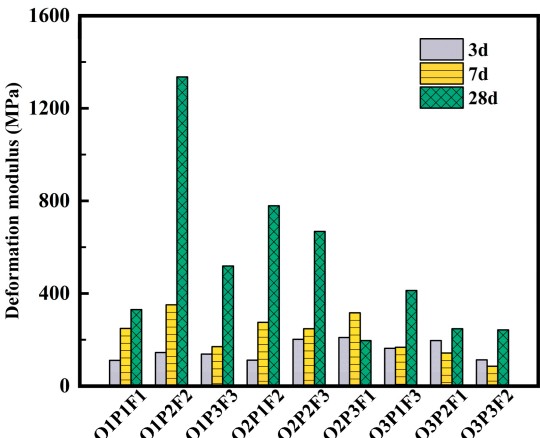

**Figure 7.** Deformation modulus of solidified soil samples.

The research in this paper still has the following deficiencies that need to be further improved:

In this paper, the mechanical properties of solidified soil from drilling waste mud are mainly studied through macro tests. There is a lack of theoretical support for micro aspects, such as the crack morphology of solidified soil, and micro aspects, such as the morphology of hydration products. Therefore, it is particularly necessary to carry out SEM tests, CT scanning tests, and XRD tests on solidified soil.

Due to the limited conditions, the mechanical research on the solidified soil of drilling waste mud only stays in the laboratory test. It is necessary to further conduct field tests on actual projects to facilitate the further combination of practical problems in the project.

This test only considers the mechanical properties of solidified soil when the moisture content of drilling mud is 50%. Drilling mud in actual engineering projects often has uncertain water content, so the influence of the different initial moisture content of drilling mud needs to be considered in future work.

## 4. Conclusions

In this study, the unconfined compressive strength test of drilling waste mud solidified soil is carried out according to the orthogonal design method, and the strength of solidified soil, stress-strain relationship, and deformation modulus of solidified soil with curing age, freeze-thaw cycle times, and curing agent ratio are discussed. The main conclusions are as follows:

(1) In the orthogonal test, when the ratio of the curing agent is O1P2F2, that is the content of cement, desulphurized petroleum coke slag, and fly ash is 8%, 22.5%, and 16%, respectively. The strength of solidified soil is the highest at any curing age, and the highest strength at 28 days is 1792.3 kPa. Through the analysis of range and variance, the primary and secondary order of curing agent components affecting the strength was obtained, and the optimal level ratio, namely O1P2F2, was determined.

(2) With the increase of the curing age, the unconfined compressive strength of solidified soil samples under each ratio increases and becomes stable after 28 days. Compared with the 3 days strength, the growth rates of 7 days strength and 28 days strength are roughly between 62–107% and 330–738%, respectively.

(3) With the application of the freeze-thaw cycle, the unconfined compressive strength of solidified soil samples shows a weakening trend. After 10 freeze-thaw cycles, the strength attenuation range is between 16.7% and 21.9%. When the number of cycles is 0–4 times, the decrease in strength reaches the largest amount, accounting for about 59.1–67.7% of the total attenuation.

(4) With the increase of curing age, the stress-strain curve of the sample becomes steeper and steeper and has a certain plasticity at 3 days. The plasticity is then gradually replaced by brittleness, and, at 28 days, it shows complete brittleness. With the increase of curing age, the peak strain of the solidified soil samples becomes smaller and smaller, from 2.5–3.0% in 3 days to 0.75–1.0% in 28 days.

(5) The deformation modulus $E_{50}$ of solidified soil increases with the increase of the curing age. At 28 days, the deformation modulus of the sample with O1P2F2 is the highest, indicating that the resistance to deformation reaches the strongest point at this time.

**Author Contributions:** Y.J. is responsible for preparing the manuscript draft and conceptualizing the research tasks and objective and is also responsible for the supervision of H.L.; H.L. and Y.M. are responsible for conducting the experiments and analyzing results; N.L. is responsible for the editing of the manuscript and re-analyzing the results. All authors have read and agreed to the published version of the manuscript.

**Funding:** This research was funded by the project Offshore Wind Farm Pile Foundation supported by Shandong Province Natural Science (ZR2021YQ31), National Natural Science Foundation of China (42277135).

**Data Availability Statement:** Data are available upon reasonable request.

**Conflicts of Interest:** The authors declare that there is no conflict of interest with anyone.

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
