# Peer review of "Study on Improving the Water Quality of Drilling Mud Using Industrial Waste Residue"

_water, doi:10.3390/w15010174_

Round 1
Reviewer 1 Report
This manuscript presents an experimental study of solidifying drilling mud waste for use in roadbed construction. The laboratory tests include UCS and free-thaw. The test program is well-designed. Detailed comments are provided below.
1. The experiment study has nothing to do with the impact of water quality. The title does not reflect the manuscript. The title can be revised as "Solidification of waste mud for road construction."
2. The choice of desulfurized petroleum coke residue as one of the agents to solidify drilling mud is not explained.
3. How is drilling mud waste stored? What is the moisture condition of the drilling mud at the site condition? In order to adopt the recommended treatment method, the drilling mud has to be in certain moisture conditions. If the drilling mud is in a slurry state, it will not be possible to treat it with the recommended solidification method. It has to be dried first, which is time-consuming and labor-intensive. A discussion of the practical application of this solidification treatment is needed.
4. Many figures are in low resolution with blurred fonts, e.g. Figure 3.
5. Drilling mud has the main component, bentonite. However, PI of the mud sample is only 22.1. what is the weight percentage of benonite in the mud sample?
6. 2.2.1 preparation of solidified soil: moisture content of the soil samples was not reported.
7. It is unclear what citation style is followed in the manuscript. The citation style needs to be checked.
Author Response
On behalf of all the authors, I would like to express my heartfelt thanks to your letter and the reviewers for their constructive comments on our article. These comments are valuable and help to improve our article. We have extensively revised the manuscript. In this revision, changes to our manuscript are highlighted in the document with red text. The point by point replies to reviewers are listed below.
- Thank you for your suggestions. We have revised the title of the article, and the new title is Study on improving the water quality of drilling mud using industrial waste residue
- Thank you for your questions and suggestions. The petroleum coke desulfuration residues used in the test is the combustion product of CFB boiler in petroleum refinery. It belongs to industrial waste. Based on the principle of environmental protection, we recycle the industrial waste from the petroleum industry as the material for solidification of drilling mud. In the last paragraph of the introduction, we added the reason for choosing desulfurization petroleum coke residue as the curing material.
- We sincerely thank the reviewers for their careful reading. Due to the limitations of this test, the drilling mud retrieved from the site was not treated as necessary. The mud moisture content measured before the test is 48.4%. Therefore, we calculate the required dried mud and water based on the moisture content of 50%. This test is conducted under the condition of 50% moisture content of mud. However, the reviewer's suggestions are very correct and necessary. We are willing to explore and further study the influence of different initial moisture content of mud on the curing effect in future work.
- Thank you very much for your suggestions. We have replaced the clearer figures.
- Thank you very much for your suggestions. We carefully checked the relevant content and figures and made adjustments. Replaced the wrong figure and rewrote the relevant content.
- We have added relevant contents of mud moisture content in this part according to the suggestions of the reviewer.
- We sincerely thank you for your valuable comments. We carefully examined the format of the references and made changes.

Reviewer 2 Report
Review Report for |
: |
WATER (MDPI) |
Manuscript # |
: |
water-2011463 |
Title |
: |
Study on the impact of water quality of the drilling waste mud |
Title:
In principle, the title is OK. However, it can still be optimized for search engines by adding a few more words to the list of 15-20 words, but not the ones that are already in the keywords. I think a good title would be: Purpose + Method. So, please consider adding "using industrial waste residue" to the current title, so that it becomes: "Study on improving the water quality of drilling mud using industrial waste residue."
Abstract:
- The existing abstract consists of 151 words. I suggest optimizing the abstract to 200 words.
- The abstract does not show the results or a brief conclusion. Abstract only displays long backgrounds, purposes, and short methods. So, I suggest adding a short explanation of the method, as well as the important results, to the conclusion.
Keywords:
Keywords are OK. However, for search engine effectiveness, please check again so that there are no duplicate keywords in the title.
INTRODUCTION:
In general, INTRODUCTION is ok and understandable, but there are some notes for improvement;
- The first sentence in the first paragraph is okay. However, please give a little more of a general overview of the opening sentence in the first paragraph.
- The second to fourth paragraphs are relatively short. After all, the topics discussed are interrelated. If there is nothing to develop further, I suggest combining the three paragraphs.
- In the literature review section, I only saw about four previous studies that were discussed. When I tried to use the keyword, "treatment of drilling waste mud using industrial residues" on Google Scholar, you can see that there are lots of studies related to your research. For that, I suggest improving the literature review section with the addition of related studies. So, the novelty of your research can be seen compared to previous studies.
- The purpose sentence in the last paragraph is quite long. I suggest separating these sentences into main purposes and explanatory sentences.
METHODOLOGY
The METHODOLOGY section has been written clearly and in detail so that it can be understood well. However, there are some notes for improvement;
- Please change section 2.1 Materials and experimental approach to 2. Materials and Methods. Please adjust the numbering of the following sections.
- I suggest that the quality of Figure 1, Figure 2, and Figure 3 should be improved.
- Please correct the writing of items in Table 1.
- Because this is the Materials and Methods section, I suggest explaining in detail the materials, instruments, and methods used.
- Are there formulas used as the basis for calculations? If there is, please show it as necessary.
RESULTS and DISCUSSION:
The information presented in RESULT and DISCUSSION is understandable. However, there are some notes for improvement;
- Please change the current section 3. Test results and analysis to 3. Results and Discussions
- I suggest that the quality of Figures 4-8 should be improved.
- The 3. The results and Discussion section should be formula free. For that, I suggest moving Equation 1 to 2. Materials and Methods section.
- The Discussion section is an important and interesting part of the paper. However, I don't see clearly the Discussion section. Maybe it's already included in the explanation in the Result section. If that is what the Authors mean, then I suggest specifically discussing some of the points to be discussed from the results that have been obtained.
- I suggest re-involving existing references in the discussion section.
- There is the last paragraph missing in the Results and Discussion section. This last paragraph explains more about; the application, contribution, and development direction of your research in the future. For that, I suggest completing and adjusting as necessary.
CONCLUSIONS:
Here are some notes for improvement in the conclusion section:
- The conclusion is too long by repeating in detail the key findings that have been presented in the Results section. I suggest staying focused on explaining 3-5 key points briefly so that readers can catch the important results of this study.
- At the end of the conclusion, you should also briefly include the items from the discussion as well as the items from the last paragraph.
REFERENCES:
The existing references are related, but not enough. I also found that there were 8 out of 12 references (75%) whose year of publication was over 5 years. For that, I suggest trying to update the latest references and also adding more related references.
ENGLISH:
I found a lot of grammatical errors from the title to the conclusion. I suggest checking English with Grammarly again and trying to maximize your Grammarly score.
Author Response
First of all, thank you very much for taking the time to read and revise my article. Thank you for your valuable advice. You have made comprehensive corrections to the structure, content, research methods and results of my paper. This has played a very important role in improving the quality of my paper.
We studied the reviewer's responses and revised the paper according to suggestions.
Title:
Thank you for your advice. We have revised the title.
Abstract:
We think this is a good suggestion. We have revised the article abstract to include a summary of important test conclusions.
Keywords
Thank you for your suggestion. We have added a keyword appropriately to your suggestion.
Introduction
We sincerely appreciate your valuable comments. We have adjusted the structure of this part of the article. We reviewed the literature carefully and added more relevant references to the introduction of the revised version.
Methodology
We have already revised the title on the recommendation of the reviewer. We also replaced the low quality figures and checked the writing of Table 1. We believe that this section mainly explains the materials, test instruments and methods used and does not involve data processing, so we do not show any formulas. We thank you very much for your suggestion.
Results and Discussion
We sincerely appreciate your suggestions. We have modified the number and content of the title. We replaced the clearer picture. We believe that the formula presented in the Results and Discussion section is to show the readers the method of data processing. Your suggestion is quite correct. However, Formula 1 is about the deformation modulus, which is the processing of test data. We do not think it is necessary to display the deformation modulus in the Material and Method part. Therefore, we still retain Formula 1 in the Results and Discussion section.
We have increased the discussion and more in-depth analysis of the test results. We added more references to support this idea.
We added a paragraph in the Results and Discussion section to explain the shortcomings of this experiment and more about the development direction of future research.
Conclusion
Thank you for your valuable suggestions. We summarized the results of the experiment into five important conclusions and summarized the main points as much as possible.
References
We sincerely thank you for your valuable comments. We have added relevant references in the past five years.
English
We did our best to improve the manuscript and made some modifications to it. These changes will not affect the content and framework of the paper.

Reviewer 3 Report
Title: “Study on the impact of water quality of the drilling waste mud”.
After reviewing the present manuscript, I found that the authors choose an important topic on the study on the impact of water quality of the drilling waste mud.
The manuscript is well written, and the presentation of data is satisfactory. The content of the manuscript is sufficient to be published in Water MDPI. However, several minor suggestions are given below. It is the reviewer's belief that these amendments will contribute to enhancing the current quality of the manuscript.
1. It is suggested for the authors to include the highlights, with the most significant findings of the review.
2. Use a full stop (.) after number in every heading/subheading.
3. Figure-3-Resolution need to be improved
4. Line 146-Indent
5. Figures 4to8- Resolution and label’s need to be improved
6. References need to be included for Macroscopic analysis, Range analysis.
Author Response
We appreciate your professional comments on our articles. As you are concerned, there are several issues that need to be addressed. Based on your suggestions, we have made changes to each of your suggestions.
- We sincerely thank the reviewers for their suggestions. We have made appropriate supplements to the Results and Discussion in the article, and deeply analyzed the mechanism of the experimental phenomenon.
- We are sorry for our carelessness. In our resubmitted manuscript, the title number has been modified. Thank you for your correction.
- Thank you for your advice. We have replaced the figures in the revised manuscript.
- Thank you for your careful inspection. We are sorry for our carelessness. According to your comments, we have made corrections.
- Thank you for your suggestion. We replaced high quality figures.
- Thank you very much for your suggestion. We have carefully examined the macro- and range-analysis sections, added several paragraphs, and quoted references from recent years.
